# Transcriptional programming and T cell receptor repertoires distinguish human lung and lymph node memory T cells

Nathan Schoettler [1,2]*, Cara L Hrusch[1], Kelly M Blaine[1], Anne I Sperling[1,3] & Carole Ober[2,3]

Antigen-specific memory T cells persist for years after exposure to a pathogen and provide effective recall responses. Many memory T cell subsets have been identified and differ in abundance throughout tissues. This study focused on CD4 and CD8 memory T cells from paired human lung and lung draining lymph node (LDLN) samples and identified substantial differences in the transcriptional landscape of these subsets, including higher expression of an array of innate immune receptors in lung T cells which were further validated by flow cytometry. Using T cell receptor analysis, we determined the clonal overlap between memory T cell subsets within the lung and within the LDLN, and this was greater than the clonal overlap observed between memory T cell subsets compared across tissues. Our results suggest that lung and LDLN memory T cells originate from different precursor pools, recognize distinct antigens and likely have separate roles in immune responses.

---

[1] Department of Medicine, The University of Chicago, Chicago, USA. [2] Department of Human Genetics, The University of Chicago, Chicago, USA. [3] These authors contributed equally: Anne I Sperling, Carole Ober. *email: nschoett@uchicago.edu

The hallmark of the adaptive immune system is the persistence of antigen-specific and long-lived memory cells capable of mounting recall responses. A number of CD4 and CD8 memory T cell subsets have been described and are distinguished by cell phenotypes and responses after peptide:MHC activation. Memory T cells are typically classified into effector memory (EM) or central memory (CM) populations. EM T cells are characterized by the ability to secrete cytokines and the capacity for cytotoxic activity[1,2], whereas CM T cells lack effector function but rapidly proliferate after stimulation and can activate other immune cell types and differentiate into EMs. More recently, tissue resident memory (TRM) T cells have been recognized as a separate subset that are able to quickly proliferate outside secondary lymphoid organs, and provide the most potent recall protection[3–5]. Thus, an array of responses is mounted by activation of memory T cell subsets that generate expanded populations of antigen-specific T cells with specialized functions.

Memory T cell subsets are also present at varying frequencies in different anatomical sites. While CMs and EMs are present in the blood and tissues, TRMs are rare in the blood but constitute the predominant memory T cell subset in many tissues[6,7]. Not surprisingly, T cell subsets also differ in their patterns of circulation with EMs trafficking through tissues, blood and secondary lymphoid organs, CMs circulating in the blood and secondary lymphoid organs, and TRMs residing mainly in tissues. However, it is becoming increasingly clear that memory T cell subsets may themselves be influenced by their local environment, and it is possible that phenotypically identical cells at different sites have independent ontogeny, antigen specificity and functional capacity.

In this study, we investigated the extent to which human T cell subsets from the lung and LDLN, two sites in close proximity but with different cellular compositions, share transcriptional profiles. Mucosal immunity imparted by T cell memory is particularly relevant at these sites given the exposure to inhaled pathogens and other stimuli. We found that a large number of genes are differentially expressed between phenotypically identical memory T cell subsets in the lung and LDLN and that there is limited clonal overlap of T cell receptor (TCR) repertoires between specific memory subsets in the lung and LDLN. These findings are consistent with lung and LDLN memory T cells recognizing different antigens and being poised for distinct responses after activation.

## Results

**Phenotypic analysis of lung and LDLN T cell subsets**. We first evaluated the frequency of memory T cells in paired lung and LDLN T cell subsets from 11 human donors using an 11-color antibody panel (Supplementary Table 1) and a traditional gating strategy (Supplementary Fig. 1) that allowed phenotyping of CD4 and CD8 naive, CM, EM and TRM subsets in each tissue. The relative proportions of these subsets varied between donors (Fig. 1a). Comparisons between tissues revealed that the frequencies of CD4 TRMs, CD4 EMs and CD8 TRMs were higher in the lung than in the LDLN (P value = 0.0069, P value = 0.0013 and P value = 0.0018, respectively), and CD4 and CD8 naive cells were higher in the LDLN than in the lung (P value = 0.0004 and P value = 0.0283, respectively; Fig. 1a, b and Supplementary Fig. 2). No differences were observed in tissue-specific frequencies of CD4 CM, CD8 EM and CD8 CM subsets. The ratio of CD4 to CD8 T cells was also higher in the LDLN than in the lung (Supplementary Fig. 2). Because CD4 and CD8 TRMs have been described as populations of non-circulating T cells with tissue specific localization[4,5], we were surprised to observe appreciable

numbers of cells with CD4 and CD8 TRM phenotypes in LDLNs (Fig. 1a, b).

To further characterize the phenotypic landscape of lung and LDLN T cells and identify discrete clusters of cells, we applied t-distributed stochastic neighbor embedding (t-SNE) to the multi-parameter cytometry data. The majority of clusters were comprised of cells from all donors, indicating that most T cell subsets are shared between individuals (Fig. 1c). In contrast, each cluster was composed predominantly of cells from either the lung or LDLN, although cells from the non-dominant tissue were interspersed in every cluster (Fig. 1d). Cells in each cluster were also overwhelmingly either of naive or memory phenotypes based on expression levels of the cell surface markers CD45RA and CD45RO, respectively (Fig. 1e). Additionally, clusters were separated into those with cells expressing either CD4 or CD8 (Fig. 1f, g). Expression of CD69, a marker of TRMs, as well as other cell markers used for phenotyping (CCR7, CD11a, CD11b, CD103, and CD169), had more variable patterns (Fig. 1h and Supplementary Fig. 3). In summary, our analyses identified phenotypically identical memory T cell subsets in the lung tissue and the LDLNs using both a traditional gating strategy and an unsupervised approach.

**Transcriptional programs in lung and LDLN T cell subsets**. The presence of memory CD4 and CD8 subsets in both the LDLN and lung raised the question of whether memory subsets have identical transcriptional programming between these two sites. To address this question, we sorted CD4 and CD8 EM, CM and TRM T cell subsets from paired lung and LDLN for RNA sequencing (Supplementary Table 2, Supplementary Data 1 and Supplementary Fig. 1). 128 samples that passed QC (see Methods) were analyzed. As expected, CD4 and CD8 T cells at each site expressed high levels of either CD4 or CD8A, respectively (Supplementary Fig. 4). At a false discovery rate (FDR) of 5%, 128 genes were differentially expressed between lung CD4 TRMs and lung CD8 TRMs and 805 genes were differentially expressed between LDLN CD4 TRMs and LDLN CD8 TRMs. As expected, CD4, CD8A, and CD8B were the most differentially expressed genes between these two cell types from each site (Fig. 2).

Principal component analysis of the RNA sequences was performed to assess patterns of gene expression between tissues and samples. Surprisingly, however, the primary clustering along principal component 1 was based on tissue of origin and not by T cell type, subset or donor (Fig. 3 and Supplementary Fig. 5). Consistent with this clustering, we observed a large number of genes that were differentially expressed between phenotypically identical T cell subsets residing in the lung compared to the LDLN: 418 genes were differentially expressed between lung CD4 TRMs and LDLN CD4 TRMs, and 1,363 genes were differentially expressed between lung CD8 TRMs and LDLN CD8 TRMs (Table 1, Fig. 4a, b). In fact, differences in gene expression between lung and LDLN memory T cells were present in all memory subsets (Fig. 4c, d, Supplementary Fig. 6 and Supplementary Data 2–6). To more formally assess the effects of tissue and cell phenotype on gene expression, we compared the number of genes differentially expressed between tissues to the number of genes differentially expressed within tissues for 5 memory T cell subsets. Phenotypically identical subsets between tissues had a higher number of genes differentially expressed compared to subsets in the same tissue (Wilcoxon rank-sum test P value < 0.01). These findings demonstrate that not only do lung and LDLNs have different proportions of memory T cell subsets, as previously shown[7], but phenotypically identical memory T cells from the lung and LDLN have different transcriptional programming.

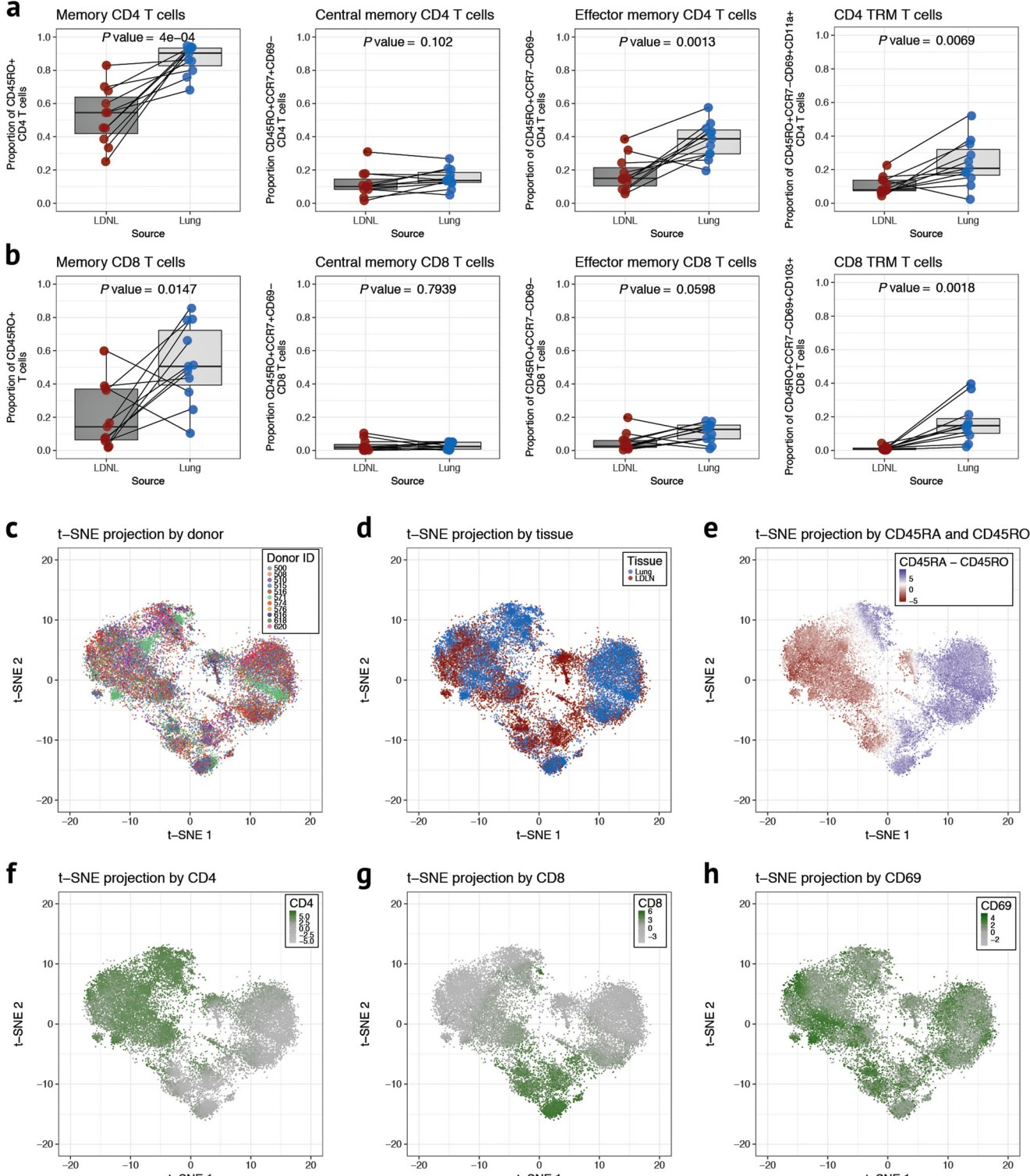

**Fig. 1** T cell phenotypes from human lung and LDLNs. **a** Cell proportions of CD4 memory, CM, EM and TRM T cells in paired lung (blue) and LDLN (red) samples (*n* = 11, paired samples from the same donor are connected by a black line) using a standard gating strategy. Proportions are relative to CD4 gated events. *P* values are from a paired t-test. Horizontal lines in the boxplot indicate median values and 25th and 75th percentile of values. **b** Cell proportions of CD8 T cell subsets as described in **a**. **c** t-SNE projection generated using cell phenotype data after random downsampling to 2500 cells for each sample with cells colored according to donor. **d** t-SNE projection with cells colored according to tissue site (lung in blue and LDLN in red). **e** Cells in t-SNE plot are colored according to the difference in the levels of CD45RA and CD45RO. t-SNE projections for levels of CD4 (**f**), CD8 (**g**) and CD69 (**h**)

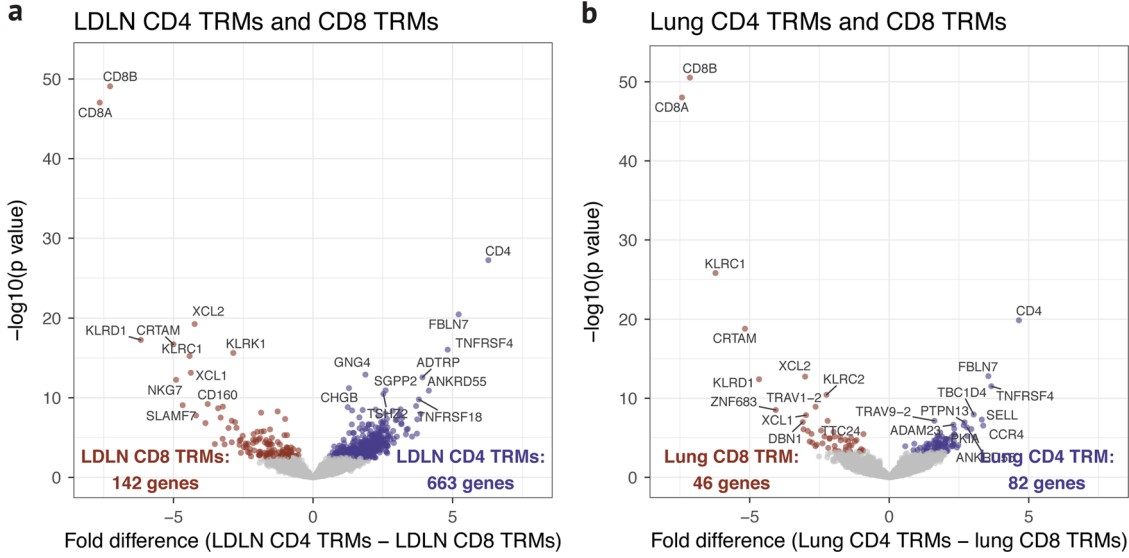

**Fig. 2** Gene expression differences between CD4 TRMs and CD8 TRMs. **a** Volcano plot comparing the gene expression between sorted LDLN CD4 TRM and LDLN CD8 TRM subsets. **b** Gene expression differences between lung CD4 TRM and lung CD8 TRM subsets. Genes with a *P* value higher than the FDR threshold of 0.05 are colored gray

The majority of genes that are differentially expressed between memory T cell subsets from the lung and LDLN are not shared by other T cell subsets. Only 78 out of 801 genes that were differentially expressed by CD4 memory subsets were shared by CM, EM and TRM subsets, and CD8 EM and TRM subsets shared 127 of 1582 genes that were differentially expressed between lung and LDLN subsets (Fig. 5). Similarly, 168 out of 1813 differentially expressed genes were shared between CD4 TRM and CD8 TRM subsets. Specific examples of differentially expressed genes include higher expression of *IL2, IL10, XCL2, GZMA*, and *GZMB* in lung CD4 TRMs compared with lung CD4 EM, whereas none of these genes were differentially expressed in LDLN CD4 TRMs and EMs. Instead, the expression of other genes was higher in LDLN CD4 TRMs compared to LDLN EMs, including *CXCL13* (Fig. 4e). Similarly, lung, CD8 EMs had higher expression of *SELL* and *S1PR5* compared with lung CD8 TRMs, while LDLN, CD8 TRMs had higher expression *S1PR1* compared to CD8 EMs (Fig. 4d). Thus, the overwhelming majority of tissue associated gene expression differences are subset specific.

**Validation of gene expression results**. We next asked whether the gene expression differences between lung and LDLN cells resulted in differences in protein levels that could be detected by flow cytometry. We tested 5 surface markers whose genes were differentially expressed between the tissues: *FCGR3A* (CD16) had higher gene expression in the lung for all CD4 and CD8 memory subsets; *CD79* and *CD9* had higher gene expression for all lung CD4 memory subsets; *CXCR5* had higher gene expression in the LDLN for all CD4 memory subsets, especially in LDLN CD4 TRMs (Fig. 3a); and *CD200* had higher gene expression in the LDLN for CD4 TRMs. Flow cytometry on paired lung and LDLN samples from the same 5 donors confirmed that the effect of tissue origin on gene expression is mirrored with cell mean fluorescence intensity (MFI) for FCGR3A (CD16), CD79, CD9, and CD200 (Fig. 6). The proportion of cells positive for each of these markers mirrored results with gene expression data for most of these markers (Supplementary Fig 7). Therefore, including these additional markers to the established memory cell surface markers will allow further classification of T cell subsets that reflects lung or LDLN localization.

**Gene ontogeny of differentially expressed genes**. Gene ontogeny (GO) analyses were performed on sets of genes with differential expression shared by certain T cell subsets. Among the 78 genes differentially expressed between tissues by all CD4 memory T cell subsets, GO identified enrichment for a number of processes, and the top processes related to immune response, immune effector processes, cellular activation and leukocyte migration (Supplementary Data 7). Similarly, the 127 genes shared by CD8 EM and CD8 TRM subsets were enriched for leukocyte immunity, leukocyte activation and immune effector process (Supplementary Data 8). Gene ontogeny analysis of genes that were differentially expressed between tissues by both CD4 TRM and CD8 TRM subsets identified highly enriched pathways that were also related to immune responses, immune effector responses, immune regulation and leukocyte degranulation (Supplementary Data 9). Collectively, this GO analysis highlights that gene expression differences between lung and LDLN that are shared by memory T cell subsets result in differences in activation status, effector capacity and migration.

**Differences between lung and LDLN TRM T cells**. We focused subsequent analyses on TRMs because these cells are non-circulating, reside in the tissue and have been shown to have unique responses in a number of tissues[5,7,8]. We first defined genes with tissue-specific expression as those that were differentially expressed between the lung and LDLN for CD4 and CD8 subsets at a 5% FDR. We then performed pathway analysis separately on the 241, 177, 899, and 464 genes with tissue-specific expression in TRM subsets in the lung and LDLNs (lung CD4, LDLN CD4, lung CD8, LDLN CD8 TRMs, respectively). Genes with higher expression in lung CD4 TRMs compared to LDLN CD4 TRMs were enriched in pathways regulated by cytokines such as IFNG, TGFB1, TNF and IL-4 (Supplementary Data 10), which were also highly connected in gene networks (Fig. 7 and Supplementary Fig. 8). These networks connected genes involved in specific immune pathways such antibody binding Fc receptor family (*FCGR2A, FCGR3A, FCGR3B* and *FCER1G*), a complement receptor gene (*C5AR1*), as well as cytokines and chemokines (*IL17A, IL1B, IL1G,* CCL20, *CXCL2,* and *CXCL3*). Therefore, these genes and pathways distinguish lung CD4 TRMs from LDLN CD4 TRMs. Among the genes with lung-specific

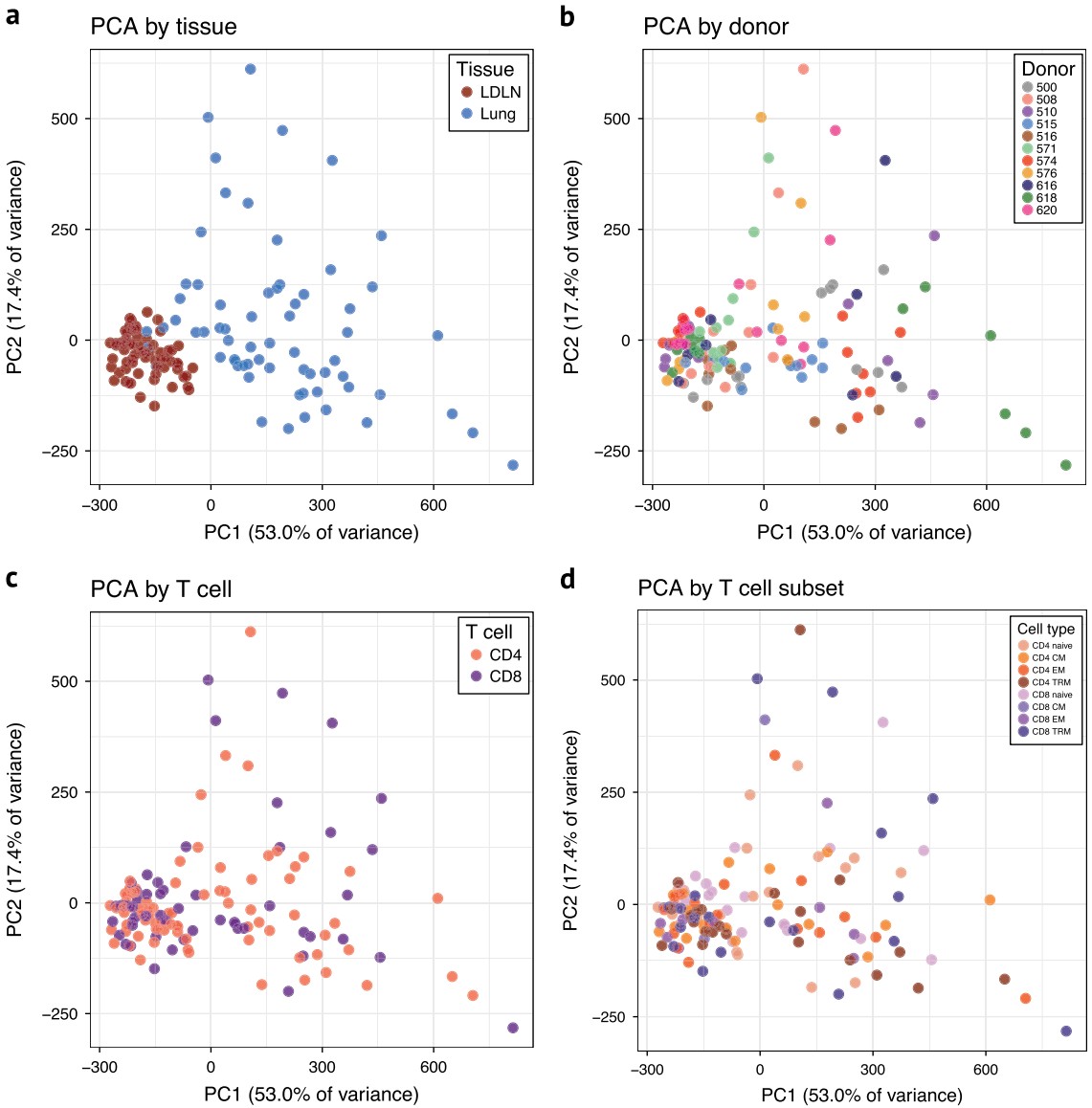

**Fig. 3** Principal component analysis of RNA sequencing data after correcting for covariates. **a** Samples are colored according to the tissue. **b** Samples are colored according to donor. **c** Samples are colored according to whether they were CD4 or CD8 subsets. **d** Samples are colored according to T cell subset

**Table 1 Number of differentially expressed genes between lung and LDLN (5% FDR)**

|         | Total | Higher expression in lung | Higher expression in LDLN |
|---------|-------|---------------------------|---------------------------|
| CD4 TRM | 418   | 241                       | 177                       |
| CD4 CM  | 235   | 197                       | 38                        |
| CD4 EM  | 346   | 304                       | 42                        |
| CD8 TRM | 1363  | 899                       | 464                       |
| CD8 EM  | 106   | 85                        | 21                        |

expression in CD8 TRMs, IFNG, TNF, IL13, TCL1A, and TGFB1 were the most promising upstream regulators. These genes formed several highly connected networks, including one with *TLR4* and *MYD88* as hubs (Fig. 7, Supplementary Fig. 7 and Supplementary Data 11). Innate sensing and signaling appears to be a prominent feature in lung CD8 TRMs, and these molecules coordinate with other differentially expressed genes, including chemokines, cytokines and *IRF3*, a transcriptional regulator of

interferon genes. Differential expression of *TLR4* and *MyD88* is not associated with classical T cell subsets, however, we also observe higher expression of one or both of these genes in lung CD4 CMs and lung CD4 EMs compared with these populations in the LDLN (Supplementary Data 3 and 4). Overall, the transcriptional profiles of lung CD4 TRMs and lung CD8 TRMs revealed higher expression of genes involved in innate immune processes in the lung compared to phenotypically similar cells in the LDLN. These genes were also enriched for inflammatory, infectious and hematologic diseases, as well as respiratory conditions, including "inflammation of the lung", "damage of lung", "infection of respiratory tract", "severe acute respiratory distress syndrome", "fibrosis of the lung" and "lung injury" (all $P$ value $< 1 \times 10^{-8}$, Supplementary Data 12). Therefore, focusing on lung T cell subsets revealed their important relationship with lung pathologies and innate immune processes.

To assess for tissue-specific processes relevant to lung TRM subsets, we analyzed the ontogeny of genes that were differentially expressed between the lung and LDLN in CD4 TRMs but not differentially expressed in other CD4 memory subsets. The top processes associated with the 255 CD4 TRM-specific genes were

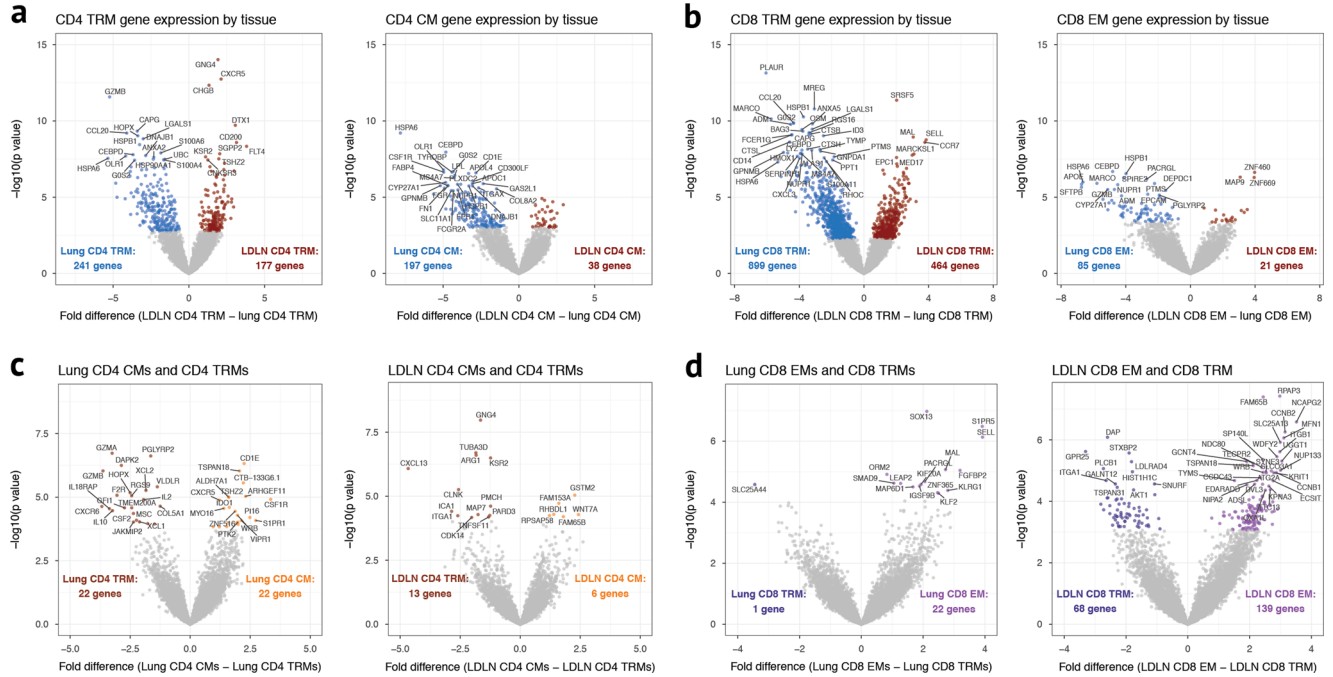

**Fig. 4** Gene expression differences between T cell subsets. **a** Comparison of gene expression between lung CD4 TRM and LDLN CD4 TRM subsets and lung CD4 CM and LDLN CD4 CM subsets. **b** Comparison of gene expression between lung CD8 TRM and LDLN CD8 TRM subsets and lung CD8 EM and LDLN CD8 EM subsets. **c** Comparison of gene expression between lung CD4 TRM and lung CD4 CM subsets and LDLN CD4 TRM and LDLN CD4 CM subsets. **d** Comparison of gene expression between lung CD8 TRM and lung CD8 EM subsets and LDLN CD8 TRM and LDLN CD8 EM subsets. See Supplementary Fig. 6 for comparison of lung CD4 EM and LDLN CD4 EM

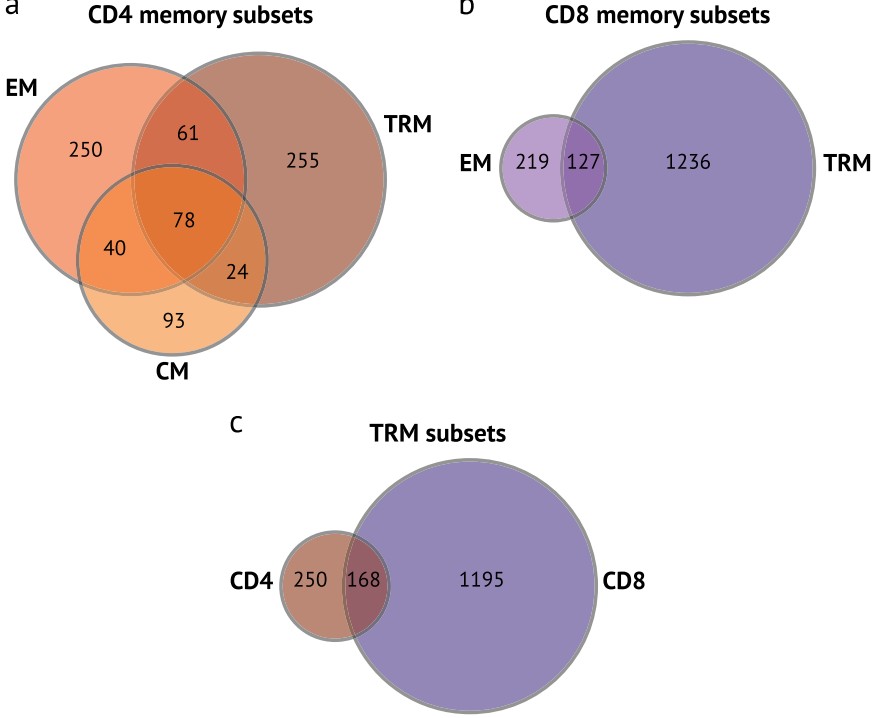

**Fig. 5** Overlap of genes differentially expressed between the tissues by memory T cell subset. Venn diagram for CD4 memory T cell subsets (**a**), CD8 memory T cell subsets (**b**) and CD4 and CD8 TRM subsets (**c**). Numbers indicate the number of genes in each section that were differentially expressed when compared between the lung and LDLN

"regulation" functions related to stress, defense, cellular processes and signaling responses (Supplementary Data 13), which contrasts from the more immune-related GO processes described above for the set of genes with differential expression shared by all memory CD4 subsets. This is consistent with CD4 TRMs playing a pivotal role in coordinating immune and inflammatory responses in a tissue-specific manner. When CD8 TRMs were similarly analyzed, the most promising GO terms were broader,

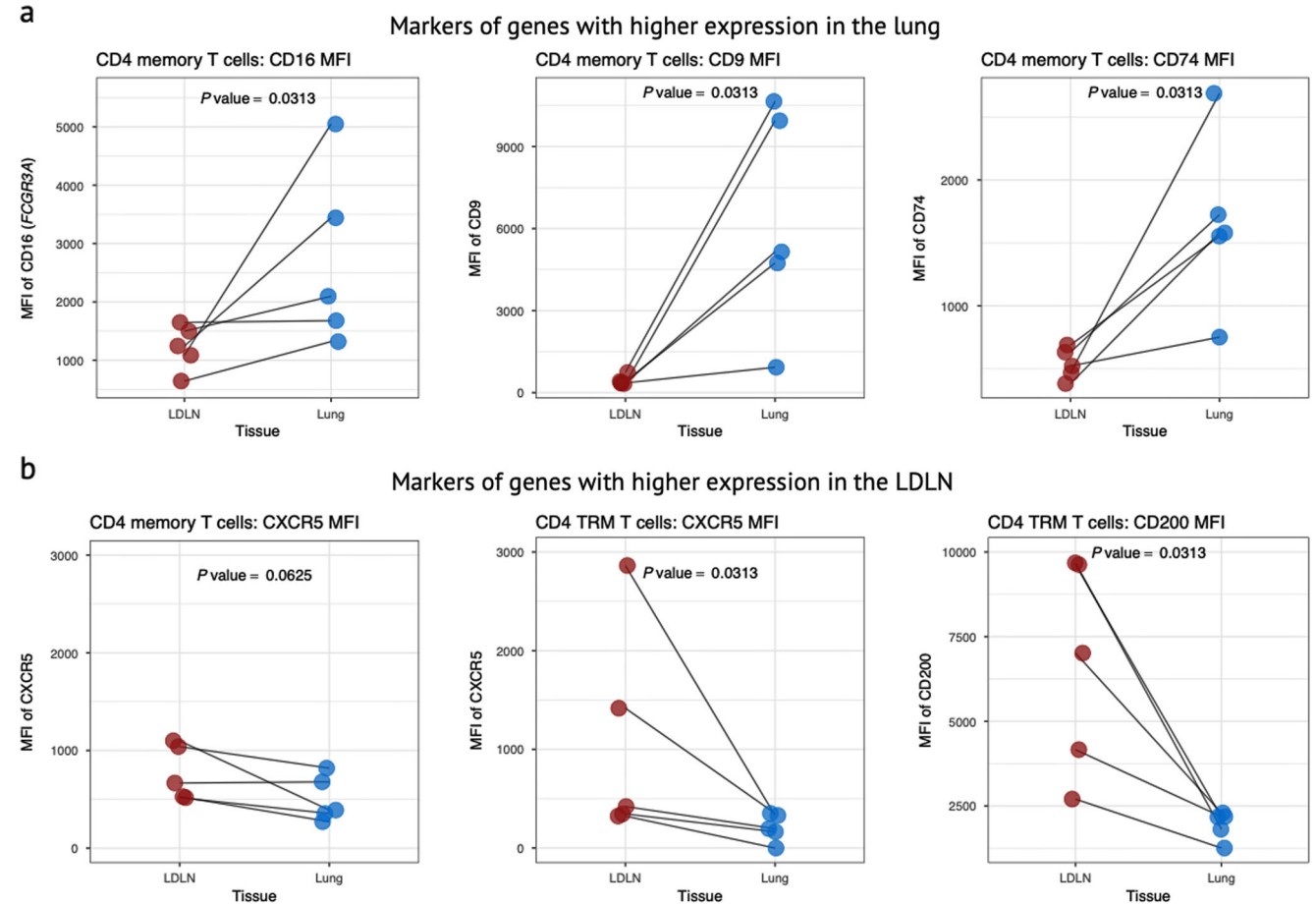

**Fig. 6** Validation of gene expression results by flow cytometry. Flow cytometry MFI for paired lung and LDLN samples from 5 donors is shown for four markers, whose genes were differentially expressed by tissue site by RNA sequencing. *P* values were calculated using Wilcoxon signed rank test for paired samples. **a** The expression of CD16, CD74 and CD9 was higher in the lung relative to the LDLN for all CD4 memory T cell subsets. The MFI is shown after gating on all CD4 memory T cells. **b** The expression levels of CXCR5 and CD200 were higher in the LDLN relative to the lung. CXCR5 gated on all memory CD4 subsets and CD4 TRMs, and for CD200 only in CD4 TRMs (gated on CD4 TRMs). MFI is shown after gating on the cells shown in each panel

including "cellular processes", "biological process", "cell activation" and "metabolic process" (Supplementary Data 14), possibly indicating that CD4 TRMs and CD8 TRMs modulate immune responses by distinct mechanisms.

**TCR repertoires of lung and LDLN memory T cell subsets.** T cell receptor specificity is determined by the recombination of TCR V, D and J gene segments that results in a vast TCR repertoire. While the TCR repertoires of memory T cell subsets are derived from the naive TCR repertoire, memory TCR repertoires are also shaped by antigen recognition and clonal proliferation resulting in a less diverse TCR repertoire memory subsets. To characterize the TCR repertoires in the sorted T cells from the lung and LDLNs, we aligned RNA sequencing reads to the TCR β chain CDR3 gene sequence. This yielded $1.12 \times 10^6$ functional (in-frame) TCR β CDR3 sequences, which was used to assess the extent of clonality within a given sample. In addition, comparisons were made of the clonal overlap between memory subsets in the lung and LDLN, and independent of clonality, whether particular V gene segments were more commonly used than others.

Our sample included 110,123 unique TCR clones. Rarefaction analysis indicated that we captured a large majority of the repertoire for nearly all samples (Supplementary Fig. 9). TCR repertoire diversity differed between samples (Supplementary

Fig. 10), but only 5.3% of the 110,123 unique clones were identified in more than one sample. We focused our analysis on the 6 of 11 donors with TCR repertoire data in at least 3 CD4 subsets from both the lung and LDLN and with more than 150 clones present in more than one subset. None of CD8 datasets from any donor met these criteria, and therefore, CD8 subsets were not analyzed (Supplementary Data 1). We first assessed the clonal overlap of CD4 memory T cell subsets from the lung and LDLN separately for each donor using clones that showed the most overlap between subsets (Fig. 8a and Supplementary Fig. 11). While several patterns of clonal overlap were observed, phenotypically identical subsets from the lung and LDLN had lower overlap than overlap of subsets within the lung or LDLN, and the most clonal overlap was between lung subsets, as indicated by more sharing of clones on the right side of the heatmaps in Fig. 8a and Supplementary Fig. 11. Next, we performed hierarchical clustering using the repertoire data from CD4 subsets for these same 6 individuals and observed clustering of subsets by tissue of origin and not CD4 subset membership (Fig. 8b and Supplementary Fig. 12). Indeed, we did not observe any instances where memory subset pairs across tissues were most closely related, and among all 6 donors, 32 of 35 memory subsets clustered according to tissue (binomial test *P* value = $4.2 \times 10^{-7}$, Fig. 8b and Supplementary Fig. 12). To further investigate clonal relationships, we applied multidimensional scaling (MDS) to the combined dataset of all subsets and all

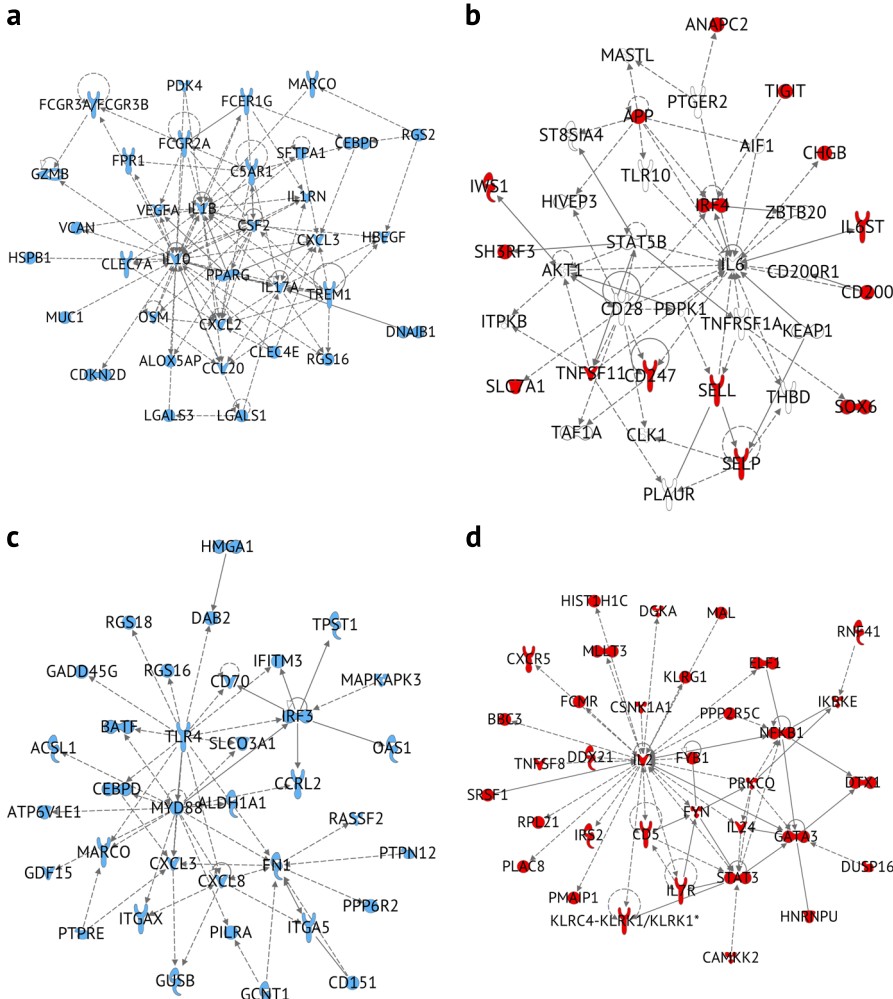

**Fig. 7** Networks of differentially expressed genes between lung and LDLN TRM subsets. **a** Network of subset of genes that had increased expression in lung CD4 TRMs compared with LDLN CD4 TRMs (IPA network score of 54). **b** Network of subset of genes that had increased expression in LDLN CD4 TRMs compared with lung CD4 TRM (network score of 21). **c** Network of subset of genes that had increased expression in lung CD8 TRM compared with LDLN CD8 TRM (network score of 37). **d** Network of subset of genes that had increased expression in LDLN CD8 TRM compared with lung CD8 TRM (network score of 57). All molecules colored (red or blue) indicate genes that were differentially expressed. Molecules without color were added to the network by the IPA software

donors (Supplementary Fig. 13). The advantage of using MDS is that distance measurements are scaled and can be quantitatively compared between all samples and individuals. Comparisons of the average pairwise distances between CD4 subsets within and between tissues, revealed low overlap (small MDS distance) between naive and memory TCR repertoires but significantly larger overlap (large MDS distance) of the TCR repertoire of memory populations in the LDLN and in the lung (Wilcoxon rank-sum test $P$ value = 0.000175 and 0.0421, respectively, Fig. 8c). While the TCR repertoire overlap between phenotypically identical T cell subsets from the LDLN and lung was larger than the overlap of naive and memory CD4 subsets (Wilcoxon rank-sum test $P$ value = 0.00354), it was not larger than the overlap of different memory T cell subsets within the LDLN or lung (Wilcoxon rank-sum test $P$ value = 0.116 and 0.879, respectively, Fig. 8c). These results show that clonal overlap of CD4 memory T cell subsets within the lung or within the LDLN is greater than the clonal overlap between phenotypically identical subsets in the lung and LDLN. Therefore, memory subsets in each of these tissues are each derived from a separate pool of progenitors with little overlap, as reflected in the modest sharing between the tissues.

We next asked whether there were particular patterns of V gene segment and J gene segment usage that occur for reasons other than clonal expansion, such as differences in V and J gene segment recombination frequencies. For this analysis, we counted each clone as a single count regardless of its frequency. Complex patterns of V gene segment and J gene segment usage were observed (shown for CD4 subsets from two individuals in Supplementary Fig. 14), but there were no prominent donor-specific patterns (Supplementary Fig. 15). We therefore pooled data from all individuals and evaluated over- or under-represented V gene or V-J gene combinations in each T cell subset. In all memory CD4 and CD8 subsets, unequal V gene usage was apparent (Fig. 9, Supplementary Fig. 15 and Supplementary Fig. 16). For example, in CD4 memory subsets from both the lung and LDLN, V5–1, V7–2, and V20–1 were over-represented (Fig. 9a, b, Supplementary Fig. 15 and Supplementary Fig. 16). These same V gene segments were overrepresented in CD8 T cell subsets from both lung and LDLN (Fig. 9c, d). Conversely, we observed that some V gene segments, including V5–2, V22–2, and V26, were rarely used in CD4 or CD8 T cells. Over and under-representation of particular V genes in memory subsets could be due to antigen selection,

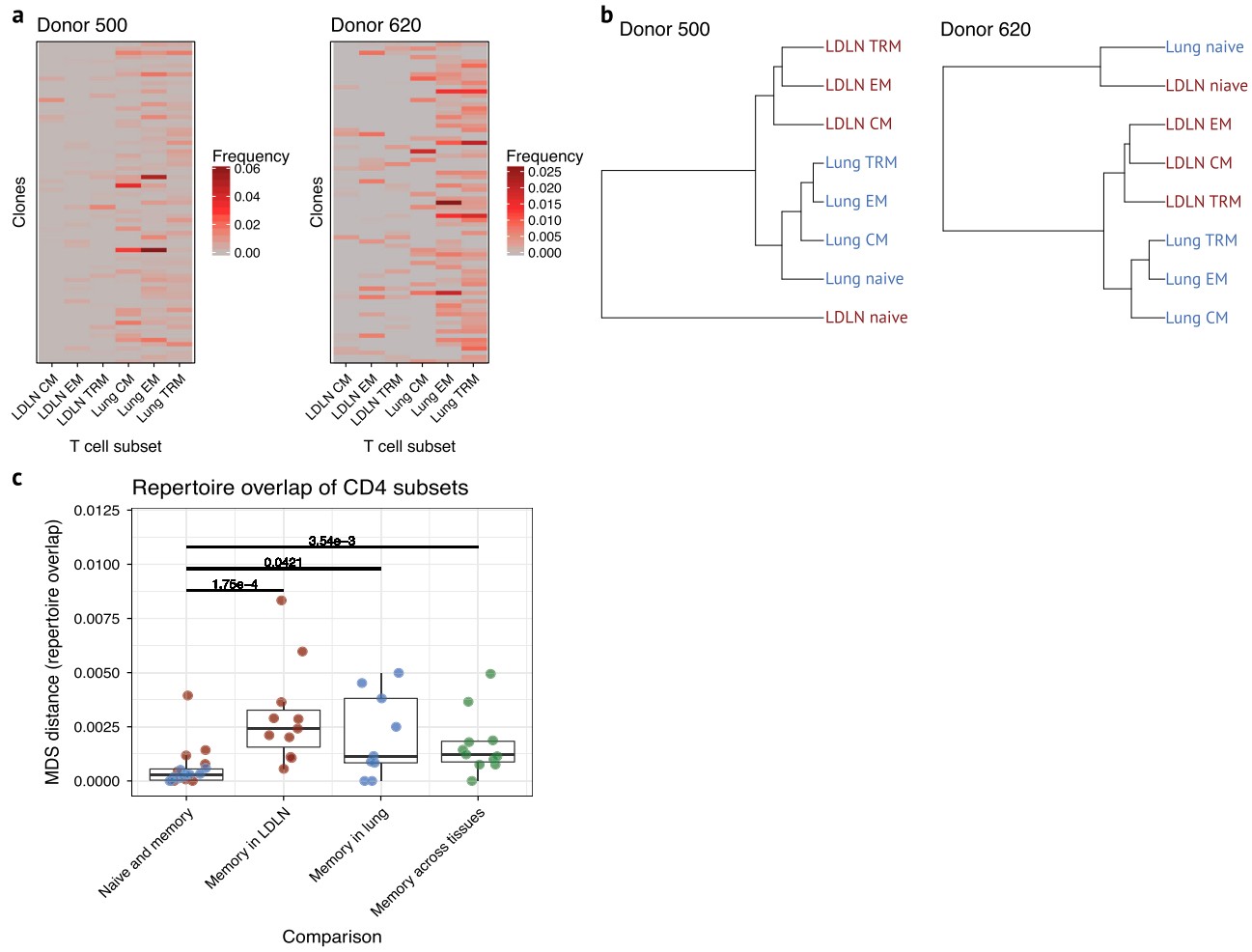

**Fig. 8** T cell receptor repertoires in lung and LDLN CD4 T cell subsets. **a** Heatmap of clonal overlap between memory CD4 T cell subsets for 2 donors. The 75 clones with the highest frequency of clonal overlap for each donor were included for this analysis. Heatmaps for other donors are shown in Supplementary Fig. 11. **b** Hierarchical clustering of CD4 subsets based on clones present in more than one subset. Clustering from two donors is shown and representative of clustering observed in other donors (shown in Supplementary Fig. 12). **c** Multidimensional scaling distance between CD4 T cell subsets for each individual is shown relative to subsets and tissue. First boxplot, distance between naive CD4 and the average of memory CD4 subsets was calculated separately for lung (blue) and LDLN (red) for each donor. Second boxplot, average of the distance between memory CD4 subsets in LDLN. Third boxplot, average of the distance between memory CD4 subsets in lung. Fourth boxplot, average of the distance between lung and LDLN memory of phenotypically identical CD4 memory subsets. Green indicates comparisons between tissues. Wilcoxon rank-sum test *P* values between groups are shown. Horizontal lines in the boxplot indicate median values and 25th and 75th percentile of values

although we observed the same patterns in naive CD4 and CD8 subsets (Fig. 9). Taken together, these observations suggest that mechanisms biasing V gene usage occur before antigen encounter and influence CD4 and CD8 T cells equivalently in the lung and LDLN.

## Discussion
In this study, we investigated the relationship between memory T cell subsets from human lungs and LDLNs and demonstrated how they differ with regard to cellular frequency, transcriptional programming, cell surface marker expression, and clonality. Our results established that memory T cell subsets are distinct between these two sites, despite the close proximity of lung tissue and LDLNs and the fact that both antigen and lung T cells drain into the LDLN. Therefore, differences between cells that are phenotypically identical using classical markers from these tissues likely result from independent roles during immune responses.

Although it is well known that lymphocyte proportions vary between tissues and typically do not reflect cell proportions in the blood, the majority of studies on human immune cell types have been conducted on whole blood or cells isolated from blood. Non-circulating memory T cell subsets, and in particular TRMs, have been reported to have tissue-specific responses and be abundant in human tissues, including the skin, gastrointestinal tract and lung[7–9]. While all of these tissues have highly specialized function, the lung's dedicated role in gas exchange necessarily juxtaposes a highly vascular pulmonary circulation with the environment. This constant exposure to the inhaled environment results in regular encounters and re-encounter with pathogens and other stimuli. In mouse models, specific lung memory T cell subsets are essential for recall responses to specific pathogens, and they are presumed to play crucial roles in humans as well[10–12]. As such, the biology of lung memory T cell subsets is highly relevant to vaccine responses and likely an array of respiratory diseases such as asthma, acute lung injury and cystic fibrosis. We have now determined that differentially expressed genes in lung memory T cell subsets are enriched in a number of inflammatory and fibrotic lung diseases as well as innate immune processes.

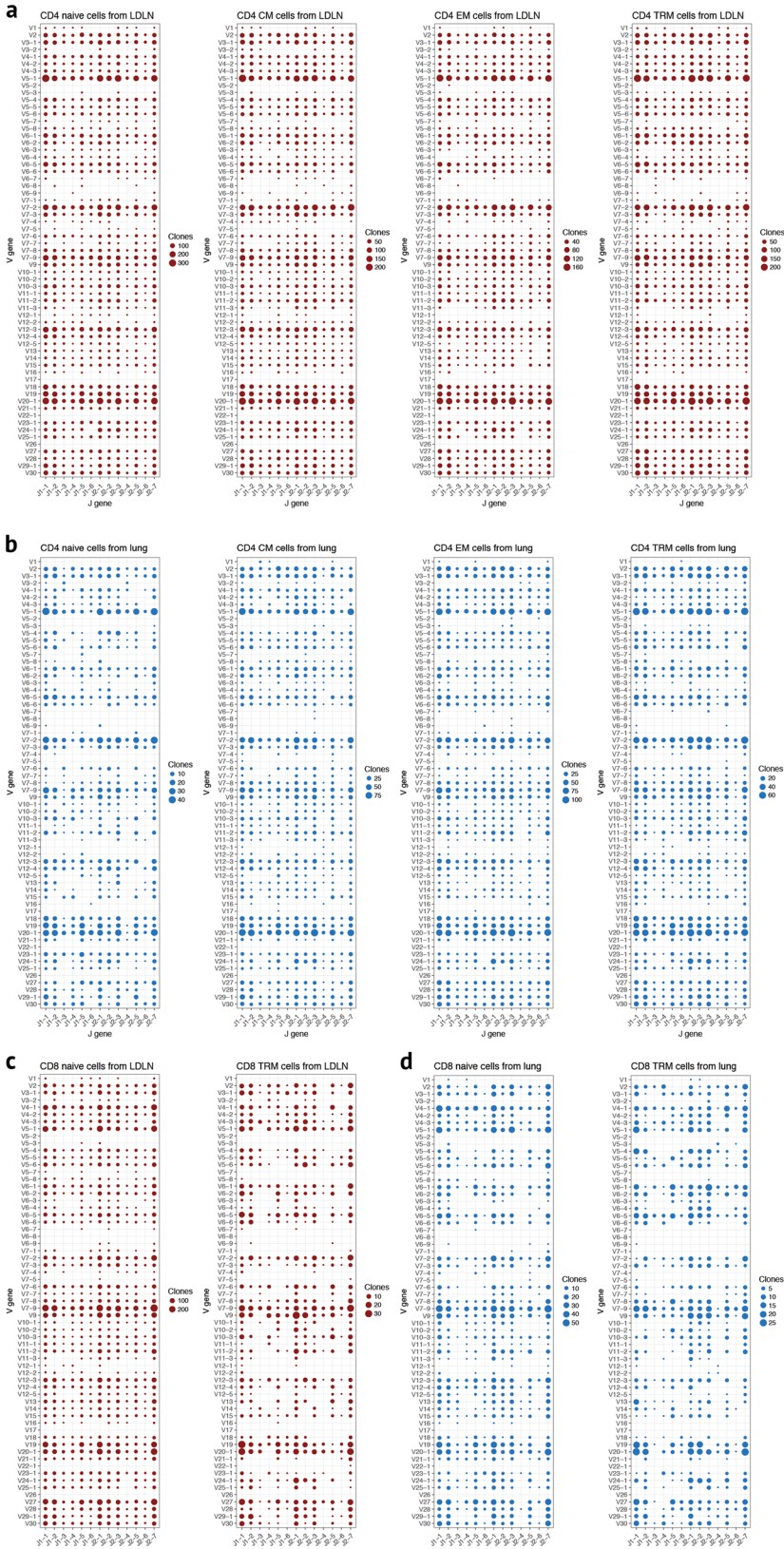

**Fig. 9** Biased V-J gene usage across subsets. Dotplots of the number of clones with a given V-J gene combination. J-gene segments are shown on the x-axis and V gene segments on the y-axis. The size of the dot is proportional to the number of clones observed with a given V-J gene recombination. V-J gene usage by CD4 T cell subset from the LDLN (**a**) and lung (**b**). CD8 naive and TRM V-J gene usage in T cells from the LDLN (**c**) and lung (**d**). Data are pooled from all 11 donors. Un-pooled data from two donors are shown in Supplementary Fig. 15

Our observations of greater proportions of CD4 and CD8 TRMs in the lung compared to the LDLN and the lower proportions of lung CMs, and particularly CD8 CMs, is consistent with previous reports[7,13]. The gene expression data described here is also consistent with other studies of CD4 and CD8 memory T cell subsets in humans. For example, *IL2* expression is higher in lung CD4 TRMs compared with lung CD4 CMs in our study, and previous studies have demonstrated high *IL2* expression in human lung and liver TRMs[14,15]. One previous study showed gene expression differences between human CD69 + and CD69- CD4 and CD8 TRM subsets, but this study was limited to analysis of three donors and to a combined analysis of blood, spleen and lung samples[14]. Another group reported gene expression differences between blood and lung CD4 + cells with the TRM phenotype[16]. Another study identified lymphoid tissue memory CD8 T cells with similar phenotypic and gene expression patterns as CD8 TRMs and these may be particularly relevant for protection against HIV[17]. In mice, gene expression differences have been reported between TRMs isolated from the spleen and TRMs isolated from the female reproductive tract after viral infection[18]. In contrast to these earlier studies, our work revealed large differences in transcriptional programs between T cells that are identical using classical phenotypes from two different sites in humans.

We confirmed that at least some of the observed transcriptional differences are biologically meaningful as all 5 of the genes we selected for further studies showed similar protein expression differences between the lung and LDLN by flow cytometry. CD9 is a tetraspanin protein that is present on many immune cells and has recently been reported to be present on memory and CD4 TRMs in the skin[19]. Increased levels of CD200 (also called OX40) on LDLN memory CD4 T TRMs plays a crucial role in costimulation of T cells. We also confirmed by flow cytometry that lung memory T cells have higher levels of proteins involved in antigen presentation, including the low-affinity Fc receptor, CD16, and the MHC class II invariant chain, CD74. Although T cells are not considered classical antigen presenting cells, induced expression of these antigen presenting surface markers in T cell subsets has been reported previously[20,21]. To our knowledge, this is the first demonstration of antigen presenting cell surface markers in human lung memory T cells.

Pathway and network analyses of genes more highly expressed in lung CD4 TRMs compared with LDLN CD4 TRMs suggest that inflammatory and regulatory cytokines are differentially expressed between these tissues, including *IL1B, IL10, IL17A, IFNG,* and *TGFB*. Compared with LDLN CD8 TRMs, lung CD8 TRMs also had higher expression of cytokine genes such as *IL10, IL6, IL17A*. Interestingly, CD8 + T cell subsets present in blood and several tissues express *IL17A*, and IL-17 secreting cells have been associated with tissue infiltration and inflammation[22]. We also observed higher expression of receptors typically associated with innate immune responses. As an example, *CD14, MYD88* and other toll like receptor genes were expressed at higher levels in lung CD8 TRMs than LDLN TRMs, and a number of innate receptors were present in lung CD4 memory subsets. While innate function is not typically attributed to classical T cell subsets, human T cells have been reported to express these genes[23], suggesting that innate sensing in the tissue may be highly relevant to these memory subsets. Our observation that gene expression of both cytokine and innate receptor genes differ between lung and LDLN subsets suggests either that these tissue-specific subsets have tremendous plasticity that is influenced by the tissue environment or that these populations are more diverse than suggested by current phenotyping by cell surface markers.

Taken together, the results of our study raise the question of whether differences in gene expression are induced in T cells by exposure to the lung and LDLN environments or whether the expression of specific genes in T cells contributes to their homing to a specific location. For example, it may be that the lung environment drives cellular characteristics of T cells. Alternatively, specific subsets may express genes and receptors that promote preferential residence in the lung. A limitation of the current study is that we cannot differentiate between these two possibilities.

We sequenced over 1 million TCRB CDR3 regions, revealing several insights about the TCR repertoires in the lung and LDLN. First, particular TCRB V gene segments are overrepresented and others are underrepresented across all T cell subsets, consistent with previous reports in other T cell subsets[24–26]. Because biased TCRB V gene usage occurred independent of clonal selection, it must be influenced by factors that affect individual V gene segment recombination and/or selection of particular gene segments during development; several of these mechanisms have been described previously[27–29]. Here, we show that the factors that globally influence V gene usage persist across a number of lung and LDLN T cell subsets, even after antigen selection into the memory pool.

Comparison of CD4 TCR repertoires revealed sharing of antigen specificity across memory CD4 T cell subsets in the lung, and the same was true of TCR repertoires of memory CD4 T cell subsets in the LDLN. It was previously demonstrated in mice that the same clone can be found in both skin TRMs and LN CMs following skin immunization[30]. However, to our knowledge, the extent of clonal overlap of human memory CD4 T cell subsets within the same tissue, and in particular between the lung and LDLN, has not been previously recognized.

While features of TCR repertoires in CD8 T cell subsets were similar to CD4 T cell subsets, we focused only on evaluating clonal overlap in CD4 T cells because some CD8 subsets were not available from most of the donors. Therefore, while it is likely that clonal overlap of CD8 T cell subsets mirrors that of CD4 T cell subsets, we did not assess this directly. A second limitation of our TCR repertoire analysis is that while we optimized our experimental strategy to recover as many cells as possible from each T cell subset, we sampled only a fraction of the overall T cells in each subset. Therefore, our analysis, while powered to identify clonal overlap between T cell subsets, does not comprehensively capture the entire TCR repertoire of each of these subsets. Additional clones are likely present in each of the subsets, and it is also possible that some of the unique clones are actually shared at low frequencies between other subset(s). Our accounting of clones also assumes that identical CDR3 nucleotide sequences arise due to clonal expansion rather than independent TRB rearrangement events. This assumption is reasonable and supported by our observation of lower clonality in naive compared with memory T cell subsets and that there is a low probability of independent TRB rearrangements generating identical CDR3 nucleotide sequences.

Models of memory T cell subset formation propose that the duration of antigen exposure and recognition drives differentiation from the naive T cell pool, with CM progeny emerging prior to the development of EM progeny[31,32]. More recently, it has been proposed that human CD8 T cell memory after yellow fever virus vaccination is maintained by quiescent cells, challenging a strictly linear differentiation[33]. In linear models of differentiation, clonal fate is determined predominantly by the antigen and TCR recognition, implying that repertoires of T cell subsets are independent. In these models, clonal overlap between T cell subsets would be low. However, we find that overlapping clones were typically shared between memory subsets in the same tissue, and therefore, a strict linear model of memory T cell differentiation does not fit with our observations. Rather, our results are

consistent with a model whereby clonal progenitors can differentiate into multiple memory T cell phenotypes, and possibly whereby residence in the lung or LDLN is imparted based on TCR specificity. The TCR repertoire findings support the observed gene expression patterns in our study, namely, that phenotypically identical T cell subsets in the lung and LDLN are independent populations.

We studied 8 different T cell subsets in human lungs and LDLN and the principal difference we identified was not whether cells were CD4 or CD8 or a particular memory subset, but rather, whether they resided in the lung or the LDLNs. This was consistent across phenotypic, transcriptional and TCR repertoire analyses. While memory T cell subsets are known to have distinct roles during immune responses, our results suggest that T cell location may be more relevant, underscoring the need for further studies of human lung T cells.

## Methods

**Human tissue procurement**. Human lung samples were obtained from organ donors whose lungs were not used for transplantation through the Gift of Hope Regional Organ Bank of Illinois. Donors with >10 pack-years of tobacco use were excluded from this study. Because samples from this study were from deceased donors, they do not qualify as "human subjects" (confirmed by the Institutional Review Board at the University of Chicago).

**Leukocyte isolation**. Cells were processed as previously described[34]. Briefly, lung tissue was perfused with sterile fetal bovine serum and phosphate-buffered saline. Paratracheal, hilar, interlobular and palpable interlobar LNs were dissected and pooled. LDLNs were mechanically dissociated into a single cell suspension and cryopreserved. The right lower lobe was minced and digested. Separately, LDLN and lung tissue mononuclear cells were enriched using density gradient centrifugation and cryopreserved.

**Cell staining and sorting**. A panel of 11 antibodies was established to discriminate leukocyte and specific T cell subsets and stained according to manufactures' recommendations. Antibody details are in Supplementary Table 1. Each paired sample (lung leukocytes and LN leukocytes from the same individual) was processed and sorted on the same day, and a minimum of two individuals had samples processed on a given day. Cells were sorted using a FACSAria Fusion (Becton Dickinson, Franklin Lakes, NJ) at the University of Chicago with the same sort template. Purity checks were performed on at least one sorted population during each day of sorting, and rarely, minor changes in compensation were made to maintain a sort purity > 98%. Cells were sorted directly into lysis buffer, immediately frozen on dry ice and stored at −80 °C.

**Flow cytometry analysis**. Flow cytometric data was analyzed using BD FACSDiva software (BD Biosciences) to determine the frequency of defined populations. Additionally, the flow cytometric data was randomly down-sampled to 2,500 CD45 + CD3 + events for each donor and analyzed using t-distributed stochastic neighbor embedding (t-SNE)[35] with the Rtsne package[36].

**Flow cytometry validation of gene expression**. Based on the results of RNA sequencing, 5 cell surface markers, CD16, CXCR5, CD9, CD74 and CD200, were chosen to be tested by flow cytometry (details on the antibodies used are shown in Supplementary Table 1). Paired lung and LDLN samples from same 5 donors used for gene expression studies were stained according to the manufacture's recommendations. Samples were run on a Fortessa X20 cytometer (Becton Dickinson). The results were analyzed using FlowJo software (Becton Dickinson) to calculate the MFI and percent of cells that were positive with gates set according to florescent-minus one controls. Memory CD4 T cells were defined as the population that was CD3+CD11b-CD4+CD8-CD45RO+CD45RA-, and similarly, memory CD8 T cells were defined as the population that was CD3+CD11b-CD8+CD4-CD45RO+CD45RA-.

**RNA sequencing**. Samples were randomized by individual, T cell subset and source (lung versus lymph node) prior to RNA extraction. RNA was extracted using All Prep DNA/RNA Mini Preparation Kit (Qiagen, Hilden, Germany). RNA quality, concentration and RNA integrity number (RIN) was assayed using an Agilent 2100 Bioanalyzer (Agilent Technologies, Santa Clara, CA). cDNA was synthesized using the SMART-Seq v4 Ultra Low Input RNA Kit (Takara Bio Company, Mountain View, CA) and libraries generated using Nextera XT DNA Library Preparation Kit (Illumina, San Diego, CA). Individual libraries were pooled and sequenced at the University of Chicago using a HiSeq 4000 (Illumina, San Diego, CA) with 100 base-pair paired-end reads. Reads were aligned to Genome

Reference Consortium Human Build 38 and gene counts were calculated using STAR 2.5.0[37]. Samples with less than 10 million mapped reads were removed. Analysis of CD8 CM subsets was not performed because fewer than 4 samples from each tissue met this threshold. Genes were filtered for those with more than one count per million in at least 10 samples, which included 14,820 genes. Principal component analysis was used to identify technical covariates.

**Statistics and reproducibility**. All statistical analyses were performed using R (version 3.3.3). Differentially expressed genes were identified using linear regression with the Limma package;[38] each donor was included as a random variable[39] while amplification cycles, age, gender and race were included as fixed variables. We controlled the false discovery rate according to the Benjamini and Hochberg method[40] with a threshold of 5%.

**Gene networks and regulators**. Ingenuity Pathway Analysis (IPA) was used to identify gene networks and regulators of genes differentially expressed in the lung and LDLN for CD4 TRM and CD9 TRM subsets. Analysis was performed using databases that included all leukocyte types, immune cell lines, bone marrow, spleen, lymph node and lung tissues.

**Gene ontology**. Gene lists were tested for enrichment in GO pathways using PANTHER version 14.2 (release 04–2018)[41,42]. The "Statistical overrepresentation test" was performed using default settings. Enrichment in the GO biological process complete annotation data set was tested using Fisher's exact test and an FDR p-value threshold of 0.05 was used.

**T cell receptor repertoire analysis**. RNA sequencing reads described above were aligned to TCR genes using MiXCR[43]. Sequences aligning to TCR β CDR3 regions that were productively rearranged (without stop codons) were included in further analysis. Clones were determined based on nucleotide sequence. Multi-dimensional scaling was calculated for all samples using clones that were present in two or more samples. For most individuals, at least one CD4 memory T cell subset was not available or did not pass quality control for sequencing. Therefore, MDS distances between memory subsets from the same individual were averaged to allow for comparison between tissues and between naive and memory subsets (Fig. 8c). For hierarchical clustering of CD4 subsets, Euclidean distances of clones present in more than one subset was calculated separately for each donor. Donors with less than 150 shared clones were excluded in this analysis.

**Reporting summary**. Further information on research design is available in the Nature Research Reporting Summary linked to this article.

## Data availability

RNA sequences generated and analyzed in the current study are available in Gene Expression Omnibus (www.ncbi.nlm.nih.gov/geo/) with an accession number GSE137967. Source data are presented in Supplementary Data.

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

## Acknowledgements

This research was supported by the National Institutes of Health Grants: NHLBI T32 HL007605, NHLBI R01 HL122712 and NIAID U19 AI095230. Human lung tissue was generously provided by the Gift of Hope Regional Organ Bank of Illinois, We appreciate contributions from Dr. Steve White and Janel Huffman to this work, and thank the following University of Chicago core facilities: Cytometry and Antibody Technology; Genomics Facility and Center for Research Informatics.

## Author contributions

N.S., C.L.H, A.I.S., and C.O. conceived the study. C.L.H. and K.M.B. collected and processed samples. N.S. conducted the experiments. N.S. and C.O. performed the data analysis. N.S., A.I.S., and C.O. contributed to manuscript writing; all authors read and approved the final version.

## Competing interests

The authors declare no competing interests.
