## [Peer Review File · Communications Biology]

Reviewers' comments:

Reviewer #1 (Remarks to the Author):

This study by Schoettler and colleagues involves analysis of T cell subsets in human lungs and lung-draining lymph nodes (LDLN) obtained from organ donors. The authors present analysis of phenotype, based on conventional T cell subset markers of naive, central memory, effector memory, and tissue resident memory T cells, and also perform whole transcriptome profiling of these subsets and extract the T cell receptor sequences from the RNAseq data. They show transcriptional differences between subsets that are tissue specific and the extent of overlap of TCR clones between the tissue site and subset. The strength of the study is in the analysis of T cells in matched pairs of lung and LN in 10-11 donors and the transcriptome profiling of multiple subsets between sites. The main weakness of the study is the lack of in-depth analysis of the differences between the subset in each site which was all based on transcriptional differences with no further investigation of the targets or pathways identified. Specific comments are below.

1. The analysis of T cell phenotype in these sites is limited to markers that delineate the major memory T cell subsets, that while it validates previous results by others, could also be taken further by linking to the transcriptome profiling. There were a number of well-known surface molecules identified in the comparison between subsets between sites that could be readily validated by flow cytometry.
2. The transcriptional analysis of the difference subsets between sites could be analyzed more fully. Figure 4 shows a comparison using volcano plots of specific subsets (TEM, TCM, TRM) between tissues and also different subsets between sites, but it went no further. Were there tissue-specific profiles identified that were common to all subsets? Were there subset specific profiles that were common across the two sites? The authors have the information needed to address these questions. Similar, the pathway analysis in Figure 5 as presented is not very clear. Were there specific nodes that were tissue or subset specific? Can these be validated? Some of the pathways included cytokines and cytokine signaling, but it is not clear how these provide insights into the maintenance of T cells in tissues or subset-specific profiles. Gene set enrichment may be a more targeted way to identify tissue- or subset-associated pathways.
3. The TCR analysis is interesting, but also a bit difficult to determine the major conclusion. There seems to be a broad range of MDS distance in the plots in Figure 6c. Do these plots combine sequences for all memory subsets? This is not clear. Could the broad differences be due to different degrees of overlap within subsets? The different VJ usage plots in Fig. 7 could be visualized more clearly maybe by using a histogram? The different sized circles are somewhat difficult to discern accurately.
4. The study could be better referenced-- some of the novelty claims may be overstated. There are several recent studies on transcriptional profiling of human lymph node memory T cells-- one in HIV infection in *Science Immunology* and the other in LDLN in *J. Immunol.* and a few others. Interestingly, many of the LN-specific transcriptional differences confirm what was found in these other studies. The overlap of memory T cells between sites has been published in previous human tissue analysis, so they are not the first to do this as claimed. While these previous studies should not diminish this current results as they are showing results in a different context and with different comparisons, it is important for the field to fully reference the other work, particularly for comparison across studies important in human work.

Reviewer #2 (Remarks to the Author):

The manuscript by Schoettler et al. demonstrates that the transcriptomes of T cells in lung-draining lymph nodes and in lungs correlate more with their tissue location than with phenotypes. These data are convincing and of interest. Based on a relative paucity of TCR CDR3 sharing between tissues, the authors conclude that the cells in lymph nodes and lungs derive from different progenitor cells. As outlined below, the repertoire data may be less conclusive than stated, which would soften the interpretation, but not distract from the main observation.

Specific comments

1. The TCR repertoire studies were based on CDR3 sequences of TRB genes in RNA-seq. It is appreciated that the authors did a rarefaction analysis, but the concern remains that RNA-seq of populations may miss infrequent specificities and bias for clonally expanded T cells, in particular since with 100,000 unique CDR3s among one million transcripts, many sequences may be singletons and sharing may be missed.
2. Clonotype definition appears to be solely based on CDR3 identity rather than TRB identity. This may be okay given the limited diversity of a memory population, but should be discussed.
3. If available, comparison of the TRB repertoires in different lung-draining lymph node samples from the same person would be of interest. Lack of sharing would imply local repertoires that are generated by on-site clonal expansion and differentiation, as suggested by Donna Farber's studies.
4. In the discussion on linear differentiation, the authors should note the recent manuscript in Nature from the Ahmed group that provide evidence for a more sequential differentiation.
5. The data on BV gene segment are of limited value.

We thank the reviewers for their excellent comments. We have responded to each comment below (in blue font) and modified the manuscript, when appropriate, which we feel is now much improved.

Reviewer #1 (Remarks to the Author):

This study by Schoettler and colleagues involves analysis of T cell subsets in human lungs and lung-draining lymph nodes (LDLN) obtained from organ donors. The authors present analysis of phenotype, based on conventional T cell subset markers of naive, central memory, effector memory, and tissue resident memory T cells, and also perform whole transcriptome profiling of these subsets and extract the T cell receptor sequences from the RNAseq data. They show transcriptional differences between subsets that are tissue specific and the extent of overlap of TCR clones between the tissue site and subset. The strength of the study is in the analysis of T cells in matched pairs of lung and LN in 10-11 donors and the transcriptome profiling of multiple subsets between sites. The main weakness of the study is the lack of in-depth analysis of the differences between the subset in each site which was all based on transcriptional differences with no further investigation of the targets or pathways identified. Specific comments are below.

1. The analysis of T cell phenotype in these sites is limited to markers that delineate the major memory T cell subsets, that while it validates previous results by others, could also be taken further by linking to the transcriptome profiling. There were a number of well-known surface molecules identified in the comparison between subsets between sites that could be readily validated by flow cytometry.

We appreciate this excellent suggestion, and conducted further flow cytometry experiments to demonstrate tissue of origin effects on protein expression. We selected 5 proteins (CD16, CXCR5, CD9, CD74 and CD200), whose gene expression levels differed between cell types of LDLN vs. lung origin, and tested for differences between protein expression and tissue origin in cells from 5 of the same donors that were used for our gene expression studies. Our results from these studies show that the gene expression differences we previously observed are mirrored by cell surface protein level differences. These markers can now be used together with well-known surface markers to identify tissue-specific memory T cell subsets. Examples of this are illustrated in the new figure 6 shown below.

We updated the manuscript to include this figure (Fig. 6) and added the following paragraph to the results (markup file lines 168-181/clean file lines 143-156).

“We next asked whether the gene expression differences between lung and LDLN cells resulted in differences in protein levels that could be detected by flow cytometry. We tested 5 surface markers whose genes were differentially expressed between the tissues: *FCGR3A* (CD16) had higher gene expression in the lung for all CD4 and CD8 memory subsets; *CD79* and *CD9* had higher gene expression for all lung CD4 memory subsets; *CXCR5* had higher gene expression in the LDLN for all CD4 memory subsets, especially in LDLN CD4 TRMs (Figure 3a); and *CD200* had higher gene expression in the LDLN for CD4 TRMs. Flow cytometry on paired lung and LDLN samples from the same 5 donors confirmed that the effect of tissue origin on gene expression is mirrored with cell mean fluorescence intensity (MFI) for *FCGR3A* (CD16), *CD79*, *CD9*, and *CD200* (Figure 6). The proportion of cells positive for each of these markers mirrored results with gene expression data for most of these markers (Supplementary Fig 7). Therefore, including these additional markers to the established memory cell surface markers will allow further classification of T cell subsets that reflects lung or LDLN localization.”

Figure 6. Validation of gene expression results by flow cytometry. Flow cytometry MFI for paired lung and LDLN samples from 5 donors is shown for four markers, whose genes were differentially expressed by tissue site by RNA sequencing. *P* values were calculated using Wilcoxon signed rank test for paired samples. **a** The expression of CD16, CD74 and CD9 was higher in the lung relative to the LDLN for all CD4 memory T cell subsets. The MFI is shown after gating on all CD4 memory T cells. **b** The expression of CXCR5 and CD200 were higher in the LDLN relative to the lung for CXCR5 in all memory CD4 subsets (left panel, gated on all CD4 memory T cells and middle panel, gated on CD4 TRMs) and for CD200 only in CD4 TRMs (right panel, gated on CD4 TRMs). MFI is shown after gating on the cells shown in each panel.

We also added the following paragraph to the discussion (markup file lines 405-416/clean file lines 332-343).

“We confirmed that at least some of the observed transcriptional differences are biologically meaningful as all 5 of the genes we selected for further studies showed similar protein expression differences between the lung and LDLN by flow cytometry. CD9 is a tetraspanin protein that is present on many immune cells and has recently been reported to be present on memory and CD4 TRMs in the skin (19). Increased levels of CD200 (also called OX40) on LDLN memory CD4 T TRMs plays a crucial role in costimulation of T cells. We also confirmed by flow cytometry that lung memory T cells have higher levels of proteins involved in antigen presentation, including the low-affinity Fc receptor, CD16, and the MHC class II invariant chain, CD74. Although T cells are not considered classical antigen presenting cells, induced expression of these antigen presenting surface markers in T cell subsets has been reported previously (20, 21). To our knowledge, this is the first demonstration of antigen presenting cell surface markers in human lung memory T cells.”

Additional details about this analysis is also included in the methods section (markup file lines 546-558/clean file lines 459-469).

2. The transcriptional analysis of the difference subsets between sites could be analyzed more fully. Figure 4

shows a comparison using volcano plots of specific subsets (TEM, TCM, TRM) between tissues and also different subsets between sites, but it went no further. Were there tissue-specific profiles identified that were common to all subsets? Were there subset specific profiles that were common across the two sites? The authors have the information needed to address these questions. Similar, the pathway analysis in Figure 5 as presented is not very clear. Were there specific nodes that were tissue or subset specific? Can these be validated? Some of the pathways included cytokines and cytokine signaling, but it is not clear how these provide insights into the maintenance of T cells in tissues or subset-specific profiles. Gene set enrichment may be a more targeted way to identify tissue- or subset-associated pathways.

We performed additional analyses as the reviewer suggested. In these revised analyses, we focused on genes that were differentially expressed between the lung and LDLN for each subset and compared genes that were common to CD4 memory subsets and common to CD8 memory subsets, as well as those that were specific to a particular subset. Fig. 5 was added to the manuscript to illustrate the overlap of differentially expressed genes between tissues and subsets. The following was also added (markup file lines 140-147/clean file lines 128-132):

“The majority of genes that are differentially expressed between memory T cell subsets from the lung and LDLN are not shared by other T cell subsets. Only 78 out of a total of 801 genes that were differentially expressed by CD4 memory subsets were shared by CM, EM and TRM subsets, and CD8 EM and TRM subsets shared 127 of a total of 1582 genes that were differentially expressed between lung and LDLN subsets (Fig. 5). Similarly, 168 out of a total of 1,813 differentially expressed genes were shared between CD4 TRM and CD8 TRM subsets (Fig. 5).”

We also compared the genes that were differentially expressed in the TRM subsets. These gene sets were then analyzed using gene ontology analysis (PANTHER) to determine whether there is enrichment for tissue- and subset-associated pathways (added lines 182-195 and lines 237-258 to markup file/lines 157-170 and 204-216 for clean file, supplementary tables 9-11 and 15-16). Added text to the manuscript includes:

“Gene ontology (GO) analyses were performed on sets of genes with differential expression shared by certain T cell subsets. Among the 78 genes differentially expressed between tissues by all CD4 memory T cell subsets, GO identified highly significant enrichment for a number of processes, and the top processes related to immune response, immune effector processes, cellular activation and leukocyte migration (Supplementary table 9). Similarly, the 127 genes shared by CD8 EM and CD8 TRM subsets were enriched for leukocyte immunity, leukocyte activation and immune effector process (Supplementary table 10). Gene ontology analysis of genes that were differentially expressed between tissues by both CD4 TRM and CD8 TRM subsets identified highly enriched pathways that were also related to immune responses, immune effector responses, immune regulation and leukocyte degranulation (Supplementary table 11). Collectively, this GO analysis highlights that the shared transcriptional differences between lung and LDLN of memory T cell subsets result in differences in activation status, effector capacity and migration.”

“To assess for tissue-specific processes relevant to lung TRM subsets, we analyzed the ontology of genes that were differentially expressed between the lung and LDLN in CD4 TRMs but not differentially expressed in other CD4 memory subsets. The top processes associated with the 255 CD4 TRM-specific genes were “regulation” functions related to stress, defense, cellular processes and signaling responses (Supplementary table 15), which contrasts from the more immune-related GO processes identified above when the set of genes with differential expression shared by all memory CD4 subsets was analyzed. This is consistent with CD4 TRMs playing a pivotal role in coordinating immune and inflammatory responses in a tissue-specific manner. When CD8 TRMs were similarly analyzed using GO, the top associations were broader and included “cellular processes”, “biological process”, “cell activation” and “metabolic process” (Supplementary table 16), possibly indicating that CD4 TRMs and CD8 TRMs modulate immune responses by distinct mechanisms.”

The methods section was updated to include how these analyses were conducted (markup file lines 589-593/clean file lines 500-504).

Overall, the results of these analyses are consistent with many of the points made previously in the manuscript, including the important findings that innate receptors are prominently expressed in lung T cell

subsets but not LDLN T cells, suggesting that innate sensing and responses by memory T cell subsets may be crucial to their homeostasis and activation.

3. The TCR analysis is interesting, but also a bit difficult to determine the major conclusion. There seems to be a broad range of MDS distance in the plots in Figure 6c. Do these plots combine sequences for all memory subsets? This is not clear. Could the broad differences be due to different degrees of overlap within subsets? The different VJ usage plots in Fig. 7 could be visualized more clearly maybe by using a histogram? The different sized circles are somewhat difficult to discern accurately.

We recognize the complexity of the TCR repertoire analysis and resulting data. We have revised the manuscript to more clearly indicate our analysis and conclusions with regard to the TCR repertoire (markup file lines 462-484/clean file lines 390-401; to addresses issues from this comment and Reviewer 2 #1-2, below):

Multidimensional scaling (MDS) was performed using all sequences that were present in more than one T cell subset (Supplementary Figure 12). The points in Figure 6c (Figure 9c in the revised manuscript) are the average MDS distances between the CD4 T cell subsets being compared as indicated on the x-axis. The reviewer is correct that the broad range of values is due to different degrees of overlap, which is now clarified on line 300.

While histograms are useful for plotting continuous variables, the data we are showing are not continuous. In Figure 7 (now Figure 9), we are plotting the number of counts (clones) for over 800 discrete V-J gene combinations, in each of 12 different T cell subsets. This figure was designed to convey the biased V-gene usage as indicated by the rows of large dots (more frequently used V-genes) and rows with few or no dots (V-genes that are rarely or never used in functional TCRs). We are limited in our ability to discern between relatively small differences in clone numbers because of the large number of data-points. We have represented these data in a number of other ways, including as heatmaps. We found that the plots in Figure 7 (now Figure 9) most clearly illustrate the points we are making. Specifically, when the data is presented as a heatmap, the low values appear to be zero and are actually more difficult to discern accurately (this is shown below on the left while the equivalent dotplot from Figure 7 is shown on the right). It should be noted that we also plot V-J gene usage data in other ways in Supplementary Figures 14-16.

4. The study could be better referenced-- some of the novelty claims may be overstated. There are several recent studies on transcriptional profiling of human lymph node memory T cells-- one in HIV infection in *Science immunology* and the other in LDLN in *J. Immunol.* and a few others. Interestingly, many of the LN-specific transcriptional differences confirm what was found in these other studies. The overlap of memory T cells between sites has been published in previous human tissue analysis, so they are not the first to do this as claimed. While these previous studies should not diminish this current results as they are showing results in a different context and with different comparisons, it is important for the field to fully reference the other work, particularly for comparison across studies important in human work.

We appreciate these suggestions and have included additional references and revised language as the reviewer suggested (markup file lines 396-402/clean file lines 324-331 and below). We agree that others have published on differences between T cell subsets in different tissues and acknowledge this work. As the reviewer points out, our findings build on the previous studies and reflect a different context and comparisons that are novel, specifically the surprising transcriptional differences between phenotypically identical T cell subsets isolated from human lungs and LDLNs.

“Another study identified lymphoid tissue memory CD8 T cells with similar phenotypic and gene expression patterns as CD8 TRMs and these may be particularly relevant for protection against HIV (17). In mice, gene expression differences have been reported between TRMs isolated from the spleen and TRMs isolated from the female reproductive tract after viral infection (18). Here, our work revealed for the first-time large differences in transcriptional programs between phenotypically identical T cells from two different sites in humans.”

Reviewer #2 (Remarks to the Author):

The manuscript by Schoettler et al. demonstrates that the transcriptomes of T cells in lung-draining lymph nodes and in lungs correlate more with their tissue location than with phenotypes. These data are convincing and of interest. Based on a relative paucity of TCR CDR3 sharing between tissues, the authors conclude that the cells in lymph nodes and lungs derive from different progenitor cells. As outlined below, the repertoire data may be less conclusive than stated, which would soften the interpretation, but not distract from the main observation.

Specific comments

1. The TCR repertoire studies were based on CDR3 sequences of TRB genes in RNA-seq. It is appreciated that the authors did a rarefaction analysis, but the concern remains that RNA-seq of populations may miss infrequent specificities and bias for clonally expanded T cells, in particular since with 100,000 unique CDR3s among one million transcripts, many sequences may be singletons and sharing may be missed.

We agree with the reviewer that our TCR repertoire analysis will miss infrequent clones. While the rarefaction analysis suggests that we captured much of the repertoire in our sample, we also want to acknowledge that we sampled only a portion of the total T cells in a given lung or LDLN pool. However, we focused on the TCR repertoire overlap between subsets and tissue sites, which does not rely on identifying every possible clone in the sample. We used the binomial distribution to estimate the limits of clone size detection based on random sampling of T cells; if a clone comprises more than 0.00475 (0.475%) of the repertoire, the p-value is larger than 0.95 for including at least one T cell from this clone in 1000 sampled T cells. Therefore, we have 95% confidence in being able to detect T cell clones present as more than 0.00475 of the repertoire when 1000 T cells were sorted. To clarify this, the following was added to the discussion (markup file lines 466-472/clean file lines 390-396):

“A second limitation of the TCR repertoire analysis is that while we optimized our experimental strategy to recover as many cells as possible from each T cell subset, we sampled only a fraction of the overall T cells in each subset. Therefore, our analysis, while powered to identify clonal overlap between T cell subsets, does not comprehensively capture the entire TCR repertoire of each of these subsets. Additional clones are likely present in each of the subsets, and it is also possible that some of the unique clones are actually shared at low frequencies between other subset(s).”

2. Clonotype definition appears to be solely based on CDR3 identity rather than TRB identity. This may be okay given the limited diversity of a memory population, but should be discussed.

Clones were defined based on TRB (T cell receptor beta) CDR3 sequences as determined by the MiXCR software package (Bolotin et al, 2015 Nat Methods, reference #40). This definition of a clone assumes a low probability that independent TRB rearrangement events will generate identical CDR3 nucleotide sequences (regardless of the TRB V-gene segments), and is standard in the literature (Qi et al, 2014, PNAS; Robins et al, 2009, Blood; Warren et al, 2011, Genome Res as examples). We have now added the following statement (markup file lines 472-485/clean file lines 396-401):

“Our accounting of clones also assumes that identical CDR3 nucleotide sequences arise due to clonal expansion rather than independent TRB rearrangement events. This assumption is reasonable and supported by our observation of lower clonality in naïve compared with memory T cell subsets and that there is a low probability of independent TRB rearrangements generating identical CDR3 nucleotide sequences.”

3. If available, comparison of the TRB repertoires in different lung-draining lymph node samples from the same person would be of interest. Lack of sharing would imply local repertoires that are generated by on-site clonal expansion and differentiation, as suggested by Donna Farber's studies.

Each lung-draining lymph node sample is pooled from several lymph nodes at the time of collection. Therefore, we are unable to perform such experiments with our samples. We do agree that this could be an interesting set

of future experiments. We had included this experimental detail in the Methods section (markup file lines 519-525/clean file line439).

4. In the discussion on linear differentiation, the authors should note the recent manuscript in Nature from the Ahmed group that provide evidence for a more sequential differentiation.

We appreciate the reviewer bringing this study to our attention as it further supports our findings. We have updated the discussion to include this reference and discussion of sequential differentiation (markup file lines 487-489/clean file lines 404-406).

5. The data on BV gene segment are of limited value.

We assume that "BV gene segment" is referencing "TCR β V- and J-gene segment", and appreciate any clarification if this is not the case. Analysis of the TCR repertoire can be done in a number of different ways and we have focused on two aspects: 1) the clonal overlap between subsets using TCR β CDR3 sequences (Figure 6, now Figure 8 in revised manuscript), and 2) the TCR β V-J gene usage independent of clonal expansion (Figure 7, now Figure 9 in revised manuscript). These are two distinct features of the TCR repertoire. We assume that some readers may be more interested in the clonal overlap analysis, while others may be interested in V-J gene usage, particularly in the context of the clonal analysis. Therefore, we prefer to present both analyses in this paper.

REVIEWERS' COMMENTS:

Reviewer #1 (Remarks to the Author):

The authors have addressed all concerns with additional experiments and analyses. I suggest to remove any and all statements which state "for the first time" or any claims, even if prefaced with "To our knowledge". Most journals do not allow these claims, and I concur that they should state the result and let it stand.

Reviewer #2 (Remarks to the Author):

The authors have done a thorough job to address the issues raised. I do not have further suggestions.